# Therapeutic Effect of Curcumin on Metabolic Diseases: Evidence from Clinical Studies

**DOI:** 10.3390/ijms24043323

**Published:** 2023-02-07

**Authors:** Yujiao Zeng, Yuting Luo, Lijie Wang, Kun Zhang, Jiayan Peng, Gang Fan

**Affiliations:** 1State Key Laboratory of Southwestern Chinese Medicine Resources, School of Pharmacy, Chengdu University of Traditional Chinese Medicine, Chengdu 611137, China; 2School of Ethnic Medicine, Chengdu University of Traditional Chinese Medicine, Chengdu 611137, China

**Keywords:** curcumin, diabetes, obesity, non-alcoholic fatty liver disease, clinical evidence

## Abstract

Metabolic diseases have become a serious threat to human health worldwide. It is crucial to look for effective drugs from natural products to treat metabolic diseases. Curcumin, a natural polyphenolic compound, is mainly obtained from the rhizomes of the genus *Curcuma*. In recent years, clinical trials using curcumin for the treatment of metabolic diseases have been increasing. In this review, we provide a timely and comprehensive summary of the clinical progress of curcumin in the treatment of three metabolic diseases, namely type 2 diabetes mellitus (T2DM), obesity and non-alcoholic fatty liver disease (NAFLD). The therapeutic effects and underlying mechanisms of curcumin on these three diseases are presented categorically. Accumulating clinical evidence demonstrates that curcumin has good therapeutic potential and a low number of side effects for the three metabolic diseases. It can lower blood glucose and lipid levels, improve insulin resistance and reduce inflammation and oxidative stress. Overall, curcumin may be an effective drug for the treatment of T2DM, obesity and NAFLD. However, more high-quality clinical trials are still required in the future to verify its efficacy and determine its molecular mechanisms and targets.

## 1. Introduction

Metabolic diseases, such as diabetes, obesity and non-alcoholic fatty liver disease (NAFLD), pose a huge threat in both developed and developing countries [1]. Diabetes is a widespread metabolic disease. The latest report from the International Diabetes Federation shows that 1 in 10 adults (20–79 years old) have diabetes, a total of 537 million people, and this number is expected to increase to 783 million by 2045 [2]. In addition, obesity has a serious impact on human life with approximately 2.8 million people dying each year from being overweight or obese [3]. NAFLD is one of the most common chronic liver diseases characterized by fatty infiltration of hepatocytes and can progress to life-threatening conditions, such as cirrhosis or hepatocellular carcinoma [4,5]. NAFLD affects 20–30% of adults in Western countries and 5–18% of adults in Asia [6]. Therefore, it is very important to find drugs that can effectively treat metabolic diseases.

In recent years, drugs such as sulphonylureas, statins and biguanides are commonly used to treat metabolic diseases. However, some adverse effects may occur with these drugs, such as gastrointestinal disorders, hypoglycemia and liver dysfunction [7,8,9]. Therefore, there is an urgent need to seek safe and effective therapeutic drugs from natural plants. Natural polyphenolic compounds are of great value in the treatment of metabolic diseases due to their beneficial antioxidant, hypolipidemic, hypotensive and anti-atherosclerotic properties [10]. Curcumin is an important natural polyphenolic compound. Accumulating evidence from animal experiments suggest that curcumin has beneficial effects on metabolic diseases due to its various pharmacological activities, such as hypoglycemic, antioxidant, anti-inflammatory, cardiovascular protection and hepatoprotective properties [11]. However, clinical trials must be performed to confirm its true effectiveness in humans. This review systematically summarizes the clinical progress of curcumin in the treatment of three major metabolic diseases, namely type 2 diabetes mellitus (T2DM), obesity and NAFLD. The information allows health workers to better understand the medicinal value of curcumin, thereby facilitating its scientific research and clinical translation.

## 2. Methods

We conducted a comprehensive literature search using the ISI Web of Science/PubMed/Science Direct/Google Scholar database to obtain information on the effects of curcumin on metabolic diseases. When retrieving the data, the following keywords were used: “curcumin” and (“metabolic disease”, “diabetes”, “obesity”, or “non-alcoholic fatty liver disease”) and (“clinical trials” or “human trials”). In addition, we used the website ClinicalTrials.gov to collect registered clinical trials of curcumin for metabolic diseases. After searching, we further examined the full text of the literature to determine eligibility for inclusion in this review. Editorials, conference abstracts and studies with incomplete or unavailable data were excluded.

## 3. Plant Sources and Chemical Properties of Curcumin

The genus *Curcuma* plants are the most important natural sources of curcumin, such as *Curcuma longa* L. and *Curcuma wenyujin* Y.H. Chen et C. Ling, *Curcuma phaeocaulis* Val., and *Curcuma kwangsiensis* S.G. Lee et C.F. Liang [12,13,14,15]. Curcumin was first isolated from the rhizomes of *Curcuma longa* L. in 1815 [16]. *Curcuma longa* is a perennial plant with fleshy, orange and tuberous rhizomes and is widely grown in India, China and Indonesia [17]. The content of curcumin in *Curcuma longa* is usually between 1% and 9% [18]. *Curcuma wenyujin*, *Curcuma phaeocaulis* and *Curcuma kwangsiensis* are native to China’s Sichuan, Guangxi and Zhejiang provinces, respectively [19]. The content of curcumin in the rhizomes of the three medicinal plants ranges from 0.068 to 1.720 mg/g [13,14,15]. The cost of curcumin is about 6 USD per gram. The chemical structure of curcumin and its main sources are shown in Figure 1.

Curcumin is a diketone in structure with the IUPAC name of (1E,6E)-1,7-bis(4-hydroxy-3-methoxyphenyl) hepta-1,6-diene-3,5-dione. Its molecular formula is C_21_H_2_O_6_ and molecular weight is 368.38 g/mol. Curcumin is an orange-yellow crystalline powder with a slightly bitter in taste. It is insoluble in water but readily soluble in polar solvents [20]. There are various methods for extracting curcumin. In addition to solvent extraction, acid–base extraction and enzyme extraction, some new technologies, such as microwave-assisted extraction, supercritical fluid extraction and emulsification liquid–liquid microextraction, can also be used to extract curcumin from plant materials [21,22].

## 4. Absorption, Distribution, Metabolism and Exclusion of Curcumin

It is reported that the oral bioavailability of curcumin is about 1% [23]. The low bioavailability of curcumin may be related to its absorption, distribution, metabolism and exclusion characteristics. Twelve healthy volunteers were randomly assigned to receive 10 g (n = 6) or 12 g (n = 6) curcumin. After 30 min, free curcumin was detected in plasma in only one subject with a maximum concentration (C_max_) of 50 ng/mL [24]. The results showed that curcumin was poorly absorbed even at large doses. Marczylo et al. investigated the distribution of curcumin in rat tissues. The results showed that curcumin and its metabolites were mainly distributed in the liver (3671.8 ng/g), kidney (206.8 ng/g) and heart (807.6 ng/g) [25]. In addition, curcumin and its conjugates were detected in the liver and portal circulation in 12 patients with liver metastasis who were given 0.45–3.60 g of curcumin daily for one week [26]. Curcumin is metabolized primarily in the small intestine and liver. Some gut microbiota, such as *Blautia*, *Bifidobacterium* and *Lactobacillus*, are involved in the biotransformation and degradation of curcumin mostly through demethoxylation, reduction, hydroxylation, methylation and acetylation [27,28]. In the liver, curcumin is reduced by enzymatic catalysis to dihydrocurcumin, tetrahydrocurcumin and hexahydrocurcumin, which are presented as free or conjugated curcuminoids [29]. Vareed et al. found that only a small amount of curcumin entered the peripheral circulation through the portal vein after oral administration, while most of it was excreted in the stool [30]. These studies have shown that curcumin has poor absorption, limited tissue distribution and extensive metabolism.

## 5. Clinical Evidence of Curcumin in Treating Metabolic Diseases

Clinical trials have proven that curcumin has beneficial therapeutic effects on T2DM, obesity and NAFLD. Table 1 lists the clinical trials registered at www.clinicaltrials.gov, 21 September 2022. In addition, Table 2 summarizes the published clinical trials associated with the three metabolic diseases.

### 5.1. T2DM

T2DM is the most common metabolic disease in the world characterized by abnormally elevated blood glucose levels [68]. Chronic hyperglycemia raises levels of advanced glycosylation end products (AGEs), which can cause inflammation and oxidative stress, thereby exacerbating T2DM progression [69]. In addition, most patients with T2DM are also accompanied by hyperlipidemia. By analyzing reported clinical trials, we found that curcumin can improve T2DM by lowering blood glucose levels, improving insulin resistance, reducing blood lipid levels and alleviating inflammation and oxidative stress (Figure 2).

#### 5.1.1. Curcumin Lowers Blood Glucose Levels and Improves Insulin Resistance in Patients with T2DM

Several clinical trials have shown that curcumin has a significant hypoglycemic effect in patients with T2DM. For example, in a double-blind randomized placebo-controlled clinical trial, patient taking 80 mg of nanocurcumin capsules daily for 3 months significantly reduced fasting blood glucose (FBG) and glycosylated hemoglobin A1c (HbA1c) levels in 35 patients (17 men and 18 women) with T2DM [31]. Other clinical trials have shown similar results. FBG and HbA1c levels were significantly reduced in patients with T2DM who took curcumin or nanocurcumin capsules compared with the placebo group [32,41,42,45].

Insulin resistance is one of the mechanisms leading to elevated blood glucose in patients with T2DM [70]. Therefore, improving insulin resistance is beneficial for alleviating T2DM. Two clinical studies conducted randomized double-blind controlled trials to evaluate the beneficial effects of curcumin capsules (150 mg bid) and curcumin C3 complex capsules (500 mg qd) for 3 months in patients with T2DM and discovered that curcumin significantly improved insulin resistance by reducing the insulin resistance index (HOMA-IR) [34,41]. Similarly, in a 6-month randomized, double-blind, and placebo-controlled trial, 107 patients (50 men and 57 women) with T2DM received curcumin capsules at 1500 mg/d and showed a significant reduction in the HOMA-IR [37]. Furthermore, a recent meta-analysis concluded that curcumin could significantly reduce FBG, HbA1c and HOMA-IR levels in patients with metabolic diseases, including T2DM [71]. Taken together, these results demonstrate that curcumin is effective in lowering blood glucose levels and improving insulin resistance in patients with T2DM.

#### 5.1.2. Curcumin Improves Blood Lipid Levels in Patients with T2DM

Dyslipidemia frequently occurs in patients with T2DM, including high levels of total cholesterol (TC), free fatty acids (FFA), triglycerides (TG), low-density lipoprotein cholesterol (LDL-C) and very low-density lipoprotein cholesterol (VLDL-C), as well as low levels of high-density lipoprotein cholesterol (HDL-C). Changes in these lipid-related markers increase insulin resistance and the risk of cardiovascular disease in patients with T2DM [72,73,74]. Several clinical trials have shown that curcumin can significantly improve dyslipidemia in patients with T2DM. For example, Panahi and Chuengsamarn et al. reported that administration of curcumin capsules (1500 mg/d and 1000 mg/d) significantly decreased plasma TG, TC and LDL-C levels and increased HDL-C levels in patients with T2DM [33,37]. Similarly, a recent study demonstrated a significant reduction in serum TG, TC, LDL-C and VLDL-C levels in patients with T2DM supplemented with nanocurcumin capsules (80 mg/d) for 12 weeks [45]. Likewise, T2DM patients supplemented with curcumin capsules (180 mg/d) for 6 months showed a significant reduction in TG levels [44]. In addition, serum lipoprotein esterase (LPL) is a key enzyme in lipid metabolism, and its increased activity will promote the hydrolysis of TGs [75]. Na et al. reported that continuous administration of curcumin capsules (300 mg/d) for 3 months significantly increased LPL activity and decreased TG and FFA levels in patients with T2DM [40,41].

#### 5.1.3. Curcumin Alleviates Inflammation and Oxidative Stress in Patients with T2DM

There is growing evidence that inflammation and oxidative stress are closely associated with the development of T2DM [76,77]. Tumor necrosis factor-α (TNF-α), IL-1β, IL-6, macrophage chemoattractant protein-1 (MCP-1) and high-sensitivity C-reactive protein (hs-CRP) are common proinflammatory markers, while IL-10, adiponectin and IL-4 are anti-inflammatory markers. In addition, malondialdehyde (MDA), glutathione (GSH) and superoxide dismutase (SOD) are indicators related to oxidative stress.

A clinical study found that taking 300 mg of NCB-02 (curcumin preparation) daily reduced serum IL-6, TNF-α and MDA concentrations in patients with T2DM [38]. Similarly, supplementation of curcumin capsule (1500 mg/d, 300 mg/d) in patients with T2DM significantly reduced TNF-α, IL-6, hs-CRP and leptin levels, while it significantly increased adiponectin and SOD levels, thereby improving oxidative stress and inflammatory states [32,33,40]. Twenty-five patients with T2DM treated with curcumin tablets (1000 mg/d) for 12 weeks showed a significant increase in total antioxidant capacity (TAC) and GSH and a significant decrease in MDA concentration [43]. Panahi et al. reported that supplementation with curcumin C3 complex capsules (1000 mg/d) reduced serum TNF-α and MDA concentrations and increased adiponectin levels, SOD activity and TAC in patients with T2DM [35,36]. Furthermore, administration of nanocurcumin capsules (80 mg/d) for 12 weeks in patients with T2DM resulted in a significant decrease in hs-CRP and MDA levels and a significant increase in TAC levels [45]. In summary, curcumin can significantly alleviate inflammation and oxidative stress, which is beneficial for the treatment of T2DM.

### 5.2. Obesity

Obesity, especially abdominal obesity, is closely associated with increased morbidity and mortality from cardiovascular disease [78]. Clinically, several anthropometric measures, such as body weight, hip circumference, waist circumference and body mass index (BMI), are commonly used to assess obesity. In addition, obese people are often accompanied by dyslipidemia and dysglycemia, such as elevated FBG, HbA1c, LDL-C, TG and TC levels. These abnormalities gradually cause inflammation and oxidative stress leading to cell, tissue and organ damage [79,80]. The effects of curcumin on anthropometric parameters, dysglycemia, dyslipidemia, inflammation and oxidative stress markers in obese or overweight people are reviewed below and shown in Figure 3.

#### 5.2.1. Effect of Curcumin on Anthropometric Parameters in Obese or Overweight Subjects

Some randomized controlled trials have been performed to investigate the effects of curcumin on anthropometric indicators in obese or overweight individuals. A double-blind, randomized, placebo-controlled clinical trial reported that supplementation with nanocurcumin capsules (40 mg bid) for 3 months significantly reduced body weight, BMI and appetite in 42 obese patients (23 men and 19 women) compared with the placebo group [54]. In addition, Pierro et al. found that administration of curcumin tablets complexed with phosphatidylserine for 30 days in overweight people with metabolic syndrome resulted in a significant decrease in BMI, body weight, waist circumference and body fat percentage [57]. However, several other clinical trials have yielded mixed results and found no significant effect of curcumin on anthropometric parameters [48,49]. Differences between studies may be related to the forms of curcumin, the dose used and the duration of intervention. Recently, a dose–response meta-analysis was performed to determine the effect of curcumin on body weight, BMI and waist circumference [81]. It was concluded that curcumin supplementation can significantly reduce body weight and BMI but significantly reduce waist circumference only when the dose is ≥1000 mg/d and the duration is ≥8 weeks. Moreover, Hariri et al. found that curcumin in more bioavailable and soluble forms can change anthropometric parameters [82]. Therefore, the drug form, dose and duration of the intervention may be the key factors affecting the clinical efficacy of curcumin.

#### 5.2.2. Effect of Curcumin on Glycemia and Lipid Levels in Obese or Overweight Subjects

Dysglycemia and dyslipidemia are common features of obesity [83]. A double-blind, randomized, placebo-controlled clinical trial was conducted to investigate the effect of nanocurcumin on glycemic and lipid profiles in overweight and obese patients [53]. The results showed that compared with the placebo group, administration of nanocurcumin capsules (80 mg/d) for 3 months significantly decreased the levels of FBG, HbA1c, HOMA-IR, TG, TC and LDL-C and increased the levels of HDL-C and the quantitative insulin sensitivity check index (QUICKI). Similarly, Karandish et al. reported that curcumin supplementation (500 mg/d) for 90 days in 21 obese subjects (5 men and 16 women) with prediabetes significantly reduced FBG, 2 h postprandial glucose (2hpp), HbA1c and HOMA-IR levels and significantly increased insulin sensitivity [56]. In addition, two randomized controlled trials found that compared with placebo, administration of curcumin C3 complex capsules (1000 mg/d) for 30 days in obese individuals or phytosomal curcumin tablets (800 mg qd) for 8 weeks in overweight subjects resulted in a significant reduction in TG levels but no significant influence on other lipid parameters, including TC, LDL-C and HDL-C [47,48].

As noted, the current clinical trials suggest that curcumin has a beneficial effect on glycemia levels in obese or overweight subjects. It can lower blood glucose, increase insulin sensitivity and improve insulin resistance. However, in addition to TG, the effect of curcumin on lipid levels cannot be reached with a definite conclusion due to the small number of clinical trials. In the future, more clinical trials with larger sample sizes are needed to determine its true effect on dyslipidemia in obese subjects.

#### 5.2.3. Curcumin Alleviates Inflammation and Oxidative Stress in Obese or Overweight Subjects

Obesity is often accompanied by inflammation and oxidative stress [79,80]. In a randomized, double-blind crossover trial, 30 obese subjects were randomly assigned to receive curcumin C3 complex capsules at 1 g/d (n = 15) or placebo (n = 15) for 4 weeks [50]. After a two-week washout period, each group was transferred to another four-week alternate treatment regimen. The results showed that curcumin significantly reduced serum IL-1β levels in obese subjects. In another randomized controlled trial, serum TNF-α, IL-6 and hs-CRP levels were significantly reduced in 42 obese patients (23 men and 19 women) treated with 80 mg/d nanocurcumin capsules for 3 months [53]. These two clinical trials have shown that curcumin has a good anti-inflammatory effect. In addition, Sahebkar et al. performed a randomized, double-blind, placebo-controlled, cross-over trial to evaluate the effects of curcumin on serum pro-oxidant–antioxidant balance (PAB) and found that curcumin was effective in reducing the oxidative stress burden in obese individuals [49]. Similarly, Saraf-Bank and colleagues evaluated the effects of curcumin on inflammation and oxidative stress in overweight and obese girl adolescents, and after 10 weeks of curcumin supplementation (500 mg/d), the levels of IL-6 and MDA were significantly decreased, while TAC was significantly increased [52]. Recently, a meta-analysis was performed to determine the impact of curcumin on oxidative stress markers [84]. It was concluded that curcumin can significantly increase total antioxidant capacity and decrease MDA concentration. In conclusion, curcumin has a promising potential to improve inflammation and oxidative stress in obese or overweight subjects.

### 5.3. NAFLD

NAFLD is also a common chronic metabolic disease that occurs when TG deposition in hepatocytes exceeds 5% of total liver weight/volume [85]. Multiple clinical trials have shown that curcumin can effectively treat NAFLD by improving liver function and hepatic steatosis, lowering blood lipid levels and relieving liver inflammation and oxidative stress (Figure 4).

#### 5.3.1. Curcumin Improves Liver Function and Hepatic Steatosis in Patients with NAFLD

Liver transaminases, including alanine aminotransferase (ALT) and aspartate aminotransferase (AST), are commonly used to check liver function in clinical practice [86]. The severity grade of NAFLD is determined by ultrasound with hepatic steatosis grades ranging from zero to three. In three randomized controlled trials, patients with NAFLD were treated with different doses of phospholipid curcumin capsules [5,61,66]. After 8 weeks of treatment, hepatic steatosis grade and serum AST and AST levels were significantly lower in the curcumin intervention group compared with the placebo group. Similarly, in two other clinical trials, patients with NAFLD taking amorphous curcumin preparations (500 mg/d) or phospholipid curcumin capsules (500 mg bid) showed significant reductions in AST and ALT levels, significant reductions in liver fat content and improvements in NAFLD severity [5,58]. In addition, two clinical studies by Panahi and Saberi-Karimian et al. found that supplementation with curcumin C3 complex capsules significantly improved the severity of NAFLD compared with the placebo group [62,64]. These results suggest that curcumin may improve liver function and hepatic steatosis in patients with NAFLD.

#### 5.3.2. Curcumin Lowers Blood Lipid Levels in Patients with NAFLD

Dyslipidemia is an important risk factor for NAFLD. Some clinical studies have shown that curcumin can significantly lower blood lipid levels in patients with NAFLD. For example, in a randomized controlled trial, 37 patients (19 men and 18 women) with NAFLD who took 500 mg of curcumin (an amorphous dispersion preparation) daily for 8 weeks showed significant reductions in serum TG, TC and LDL-C levels compared with the placebo group [58]. In addition, Panahi et al. reported that curcumin supplementation (500 mg bid) for 8 weeks significantly reduced LDL-C, TG, TC and non-HDL-C levels in patients with NAFLD [59]. In another clinical study, 61 patients (37 men and 24 women) with NAFLD were randomly assigned to phospholipid curcumin capsules (250 mg qd) and placebo groups. After 8 weeks of intervention, HDL-C levels in the phospholipid curcumin group were significantly higher than those in the placebo group [60]. Likewise, a single-arm, before-and-after, controlled clinical trial found that supplementation of phospholipid curcumin capsules (1500 mg/d) for 8 weeks significantly reduced LDL-C, TG and non-HDL-C levels in patients with NAFLD [67].

#### 5.3.3. Curcumin Alleviates Inflammation and Oxidative Stress in Patients with NAFLD

The pathological mechanism of NAFLD involves oxidative stress and a sustained chronic inflammatory response [87]. 8-hydroxy-2’-deoxyguanosine (8-OHdG), a potential marker of oxidation-related DNA damage, has been shown to be elevated in patients with NAFLD [88]. In addition, carboxymethyl lysine (CML) is an AGEs produced as a result of enhanced oxidative stress in NAFLD, and the interaction of AGEs with the receptor for advanced glycosylation end products (RAGE) in hepatocytes can worsen oxidative stress [89,90]. A randomized controlled trial by Mirhafez et al. showed that giving phospholipid curcumin capsules (250 mg qd) for 8 weeks significantly reduced 8-OHdG and CML levels in patients with NAFLD, suggesting that curcumin may improve oxidative stress in NAFLD [61]. In addition, three randomized controlled trials found that curcumin significantly reduced inflammation in patients with NAFLD [53,60,62]. Specifically, curcumin decreased levels of the inflammatory factors TNF-α, IL-6, hs-CRP, MCP-1 and leptin while increasing levels of the anti-inflammatory marker adiponectin.

## 6. Side Effects of Curcumin in Human Trials

Although curcumin has great potential in the treatment of T2DM, obesity and NAFLD, its side effects should also be evaluated. Several human clinical trials have reported that curcumin administration causes some mild gastrointestinal reactions, such as constipation, nausea, mild diarrhea, stomach pain and flatulence, in patients with T2DM, obesity or NAFLD (Table 2). However, these adverse events occurred only in individual patients, and no serious adverse events occurred. For example, in a clinical trial involving 107 patients with T2DM, curcumin intervention for 6 months resulted in a few mild adverse events, with constipation in only two subjects and nausea in only one subject [37]. Similarly, a double-blind, randomized, placebo-controlled clinical trial found that after 3 months of nanocurcumin intervention, almost no subjects experienced side effects during the study except for one subject with nausea [53]. In addition, some clinical studies have shown that curcumin has no side effects even at doses as high as 1500 mg/d [42,65]. These results suggest that curcumin is a safe natural product, and its side effects are mild and sporadic. Based on this property, curcumin has been approved as “Generally Recognized as Safe” (GRAS) by the United States Food and Drug Administration [91]. The safety profile of curcumin opens up a broader prospect for its clinical application in the treatment of metabolic diseases.

## 7. Conclusion and Future Prospects

It is of great significance to find effective and low toxicity drugs to treat metabolic diseases from natural products. Curcumin has been consumed as a dietary compound for centuries, and its extensive biological activities have been well tested. This paper provides an updated review of the clinical advance of curcumin in the treatment of T2DM, obesity and NAFLD. Its effects and mechanisms in humans have been classified and summarized in detail. The information may provide an important reference for the clinical application and drug development of curcumin for the treatment of the three metabolic diseases in the future.

The accumulated clinical evidence indicates that curcumin has a good therapeutic effect on T2DM, obesity and NAFLD by regulating various metabolic and pathophysiological processes. Curcumin plays a positive role in the treatment of T2DM by regulating blood glucose and lipid levels, improving insulin resistance and alleviating inflammation and oxidative stress. Similarly, curcumin is effective in the treatment of human obesity by improving anthropometric indicators (e.g., body weight and waist circumference), regulating blood glucose and lipid levels and reducing inflammation and oxidative stress. In addition, the improvement of NAFLD by curcumin is associated with its regulation of liver function and steatosis, alleviation of inflammation and oxidative stress and reduction of lipid levels.

Despite these benefits, there are still some limitations and gaps in current clinical trials of curcumin. For example, most clinical trials involved small numbers of patients with only two studies enrolling more than 100 patients. More clinical studies with larger sample sizes are needed to verify its effectiveness against T2DM, obesity and NAFLD. Second, the dose–response effect of curcumin on these three diseases has not been studied. Thirdly, some studies have produced inconsistent results, such as the effect of curcumin on anthropometric parameters in obese patients. The discrepancy may be attributed to the different delivery forms, doses and intervention duration of curcumin. Moreover, the clinical efficacy of curcumin is related to gender. Two clinical trials showed that curcumin significantly reduced glucose and LDL-C levels in male subjects with T2DM, while no significant changes were observed in female subjects with T2DM [33,34]. Therefore, more attention should be paid to the influence of these factors on the clinical efficacy of curcumin in the future. Fourthly, the therapeutic effects of curcumin in improving insulin resistance and alleviating inflammation and oxidative stress have been demonstrated in clinical trials, but the exact mechanism in humans remains unclear. Further research is needed to reveal its molecular mechanism and target in humans. Additionally, curcumin has poor oral bioavailability due to its poor solubility and pharmacokinetic properties. Therefore, improving the oral bioavailability of curcumin is a key problem to be solved in clinical application. At present, curcumin is administered in various delivery forms, such as curcumin capsules, curcumin tablets and nanocurcumin capsules (Table 2). Among them, the novel nanocurcumin formulation has a good clinical application prospect because it can improve the oral bioavailability of curcumin [92]. In addition to nanocapsules, more novel delivery systems should be investigated and applied to enhance the clinical benefits of curcumin.

In conclusion, curcumin has a high safety profile and is a promising natural product for the treatment of metabolic diseases. Further studies are needed to investigate the dose–response effects of curcumin in these three metabolic diseases and to optimize its optimal therapeutic cycle. In addition, more large-scale, long-term, high-quality clinical trials are necessary in the future to assess the efficacy and safety of curcumin in the treatment of these three metabolic disorders in order to promote its drug development and clinical application. As we gain a better grasp of the health benefits of curcumin, its therapeutic uses will become clearer and more trustworthy.

## Figures and Tables

**Figure 1 ijms-24-03323-f001:**
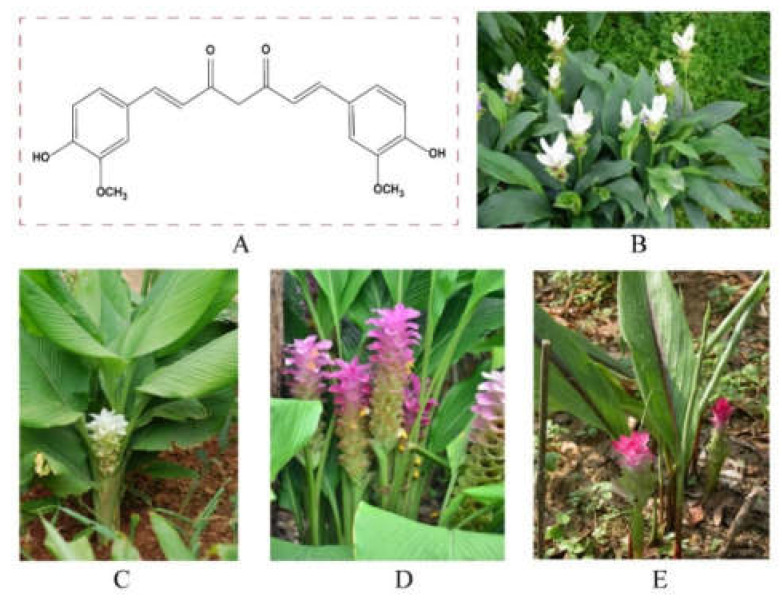
Chemical structure (**A**) of curcumin and its main plant sources ((**B**) *Curcuma wenyujin* Y.H. Chenet C. Ling; (**C**) *Curcuma longa* L.; (**D**) *Curcuma phaeocaulis* VaL.; (**E**) *Curcuma kwangsiensis* S.G. Lee et C.F. Liang). The pictures are from the website: http://www.iplant.cn/frps, accessed on 21 September 2022.

**Figure 2 ijms-24-03323-f002:**
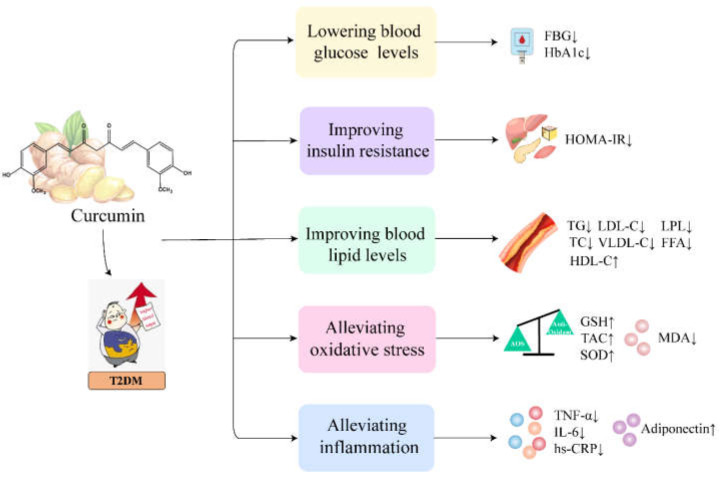
The therapeutic effects and potential mechanisms of curcumin on T2DM. ↑ indicates increase and ↓ indicates decrease. Specifically, curcumin can lower blood glucose and lipid levels and improve insulin resistance. Moreover, curcumin alleviates inflammation by decreasing the levels of IL-6, TNF-α and hs-CRP, and increasing the anti-inflammatory cytokine adiponectin. Curcumin alleviates oxidative stress by decreasing the levels of MDA and increasing the expression of antioxidants, including SOD, GSH and TAC.

**Figure 3 ijms-24-03323-f003:**
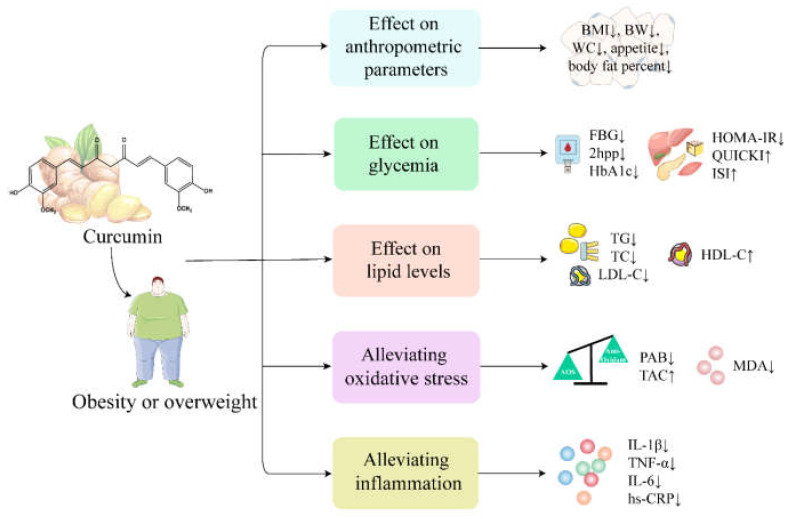
The therapeutic effects and potential mechanisms of curcumin on obesity. ↑ indicates increase and ↓ indicates decrease. Specifically, curcumin can alleviate inflammation by decreasing the levels of IL-1β, IL-6, TNF-α and hs-CRP. Curcumin alleviates oxidative stress by decreasing the levels of MDA and PAB and increasing the expression of the antioxidant’s TAC.

**Figure 4 ijms-24-03323-f004:**
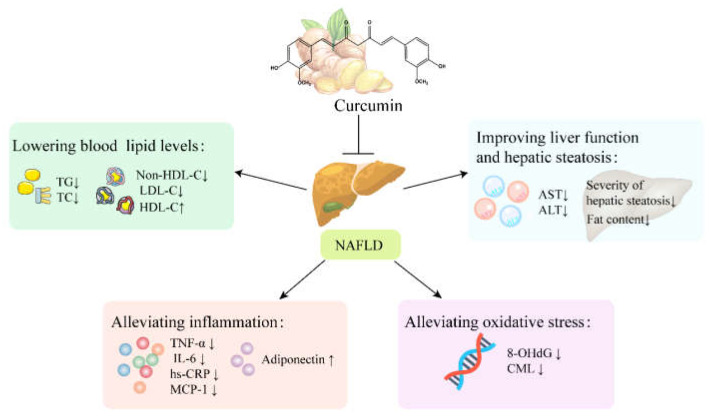
The therapeutic effects and potential mechanisms of curcumin on NAFLD. ↑ indicates increase and ↓ indicates decrease. Specifically, curcumin can improve liver function and hepatic steatosis and lower blood lipid levels. Moreover, curcumin alleviates inflammation by decreasing the levels of IL-6, TNF-α, hs-CRP and MCP-1, and increasing the anti-inflammatory cytokine adiponectin. Curcumin alleviates oxidative stress by decreasing the expression of 8-OHdG and CML.

**Table 1 ijms-24-03323-t001:** Clinical trials of curcumin in treating metabolic diseases registered at ClinicalTrials.gov.

Disease	Status	Phase I/II/III/IV	Estimated Enrollment	Intervention/Treatment	Dose	Duration	Start Date	Identifier
T2DM	-	IV	200	Curcumin capsules	500 mg tid	12 months	July 2009	NCT01052597
T2DM	-	II, III	176	Curcumin capsules	400 mg bid	26 weeks	1 August 2018	NCT03262363
T2DM	-	II, III	50	Curcumin capsules	500 mg/d	12 weeks	July 2015	NCT02529969
T2DM	Completed	-	44	Curcumin capsules	500 mg tid	10 weeks	July 2015	NCT02529982
T2DM	Recruiting	IV	60	Curcumin, glimepiride and black pepper	1100 mg/d	3 months	25 November 2020	NCT04528212
T2DM/NAFLD	Completed	II, III	50	Curcumin capsules	1500 mg qd	12 weeks	10 February 2017	NCT02908152
T2DM Pre-diabetes	-	IV	200	Curcumin capsules	500 mg tid	12 months	August 2009	NCT01052025
High cholesterol obesity	Completed	-	15	Curcumin	500 mg/d	12 months	19 September 2018	NCT03542240
Prediabetes	-	IV	142	Curcumin and bioperine	500 mg qd	3 months	25 February 2019	NCT03917784
Obesity in childhood	-	-	300	Curcumin capsules and black pepper	600 mg/d	3 months	8 January 2018	NCT03670875
NAFLD/insulin resistance	Completed	-	39	Phospholipid curcumin tablets	500 mg bid	6 weeks	5 March 2019	NCT03864783
NAFLD/insulin resistance	Recruiting	-	36	Curcumin tablets	100 mg bid	11 days	1 December 2019	NCT04315350

**Table 2 ijms-24-03323-t002:** Published clinical trials of curcumin in the treatment of metabolic diseases.

Disease	Sample Size (Test/Control)	Gender (Male/Female)	Curcumin Form	Purity	Dose	Duration	Outcome	Side Effect	Ref.
T2DM	35/35	31/39	Nano-curcumin capsules	100%	80 mg qd	3 months	HbA1c↓, FBG↓, BMI↓	No report	[31]
T2DM	21/23	22/22	Curcumin capsules	88%	500 mg tid	10 weeks	adiponectin↑, hs-CRP↓, BW↓, FBG↓, BMI↓	No report	[32]
T2DM	50/50	51/49	Curcumin C3 complex capsules	100%	1000 mg/d	12 weeks	BW↓, BMI↓, TC↓, HDL-C↑, Non-HDL-C↓, LP(a)↓	No side effects	[33]
T2DM	50/50	51/49	Curcumin C3 complex capsules	100%	500 mg qd	3 months	BW↓, BMI↓, FBG↓, HbA1c↓, C-peptide↓, AST↓, ALT↓	No side effects	[34]
T2DM	50/50	51/49	Curcumin C3 complex capsules	100%	1000 mg/d	12 weeks	TNF-α↓, leptin↓, adiponectin↑, leptin: adiponectin ratio↓	No side effects	[35]
T2DM	50/50	51/49	Curcumin C3 complex capsules	100%	1000 mg/d	12 weeks	TAC↑, SOD↑, MDA↓	No side effects	[36]
T2DM	107/106	97/116	Curcumin capsules	100%	750 mg bid	6 months	adiponectin↑, leptin↓, HOMA-IR↓, TG↓, TC↓, LDL-C↓, HDL-C↑, SUA↓, PWV↓	2 patients had constipation and 1 patient had nausea	[37]
T2DM	23/21	23/21	Curcumin capsules (NCB-02)	100%	150 mg bid	8 weeks	MDA↓, IL-6↓, TNF-α↓	2 patients had mild diarrhoea	[38]
T2DM	8	Unknown	Curcumin capsules	100%	475 mg qd	10 days	LDL-C↓, VLDL-C↓, TG↓, HDL-C↑	No side effects	[39]
T2DM with obesity	50/50	Unknown	Curcumin capsules	100%	150 mg bid	3 months	A-FABP↓, TNF-α↓, CRP↓, IL-6↓, SOD↑, FFA↓	No report	[40]
T2DM with obesity	50/50	49/51	Curcumin capsules	97.49%	150 mg bid	3 months	FBG↓, HbA1c↓, HOMA-IR↓, FFA↓, TG↓, LPL↑	No report	[41]
T2DM with obesity	21/23	22/22	Curcumin capsules	88%	500 mg tid	10 weeks	FBG↓, WC↓, BW↓	No side effects	[42]
T2DM with CHD	25/24	Unknown	Curcumin tablets	100%	1000 mg/d	12 weeks	MDA↓, TAC↑, GSH↑, PPAR-γ↑	No side effects	[43]
T2DM/IGT	15/18	22/11	Curcumin capsules	100%	90 mg bid	6 months	TG↓, γ-GTP↓	No report	[44]
T2DM on hemodialysis	26/27	32/21	Nano-curcumin capsules	100%	80 mg qd	12 weeks	FBG↓, TG↓, VLDL-C↓, FINS↓, TC↓, LDL-C↓, hs-CRP↓, MDA↓, TAC↑, PPAR-γ↑	No side effects	[45]
Prediabetes	119/116	83/152	Curcumin capsules	100%	750 mg, bid	9 months	Adiponectin↑, HOMA-IR↓, C-peptide↓, HOMA-β↑	1 patient had itching, 2 patients had constipation and 1 patient had vertigo	[46]
Overweight	40/40	Unknown	Phospholipid curcumin tablets	25%	800 mg qd	8 weeks	FINS↓, TG↓, GOT↓, GPT↓, cortisol↓	3 patients had abdominal discomfort	[47]
Obesity	15/15	5/25	Curcumin C3 complex capsules	100%	1000 mg/d	30 days	TG↓	2 patients had constipation, 1 patient had diuresis and 1 patient had paramenia	[48]
Obesity	15/15	Unknown	Curcumin C3 complex capsules	100%	1000 mg/d	30 days	PAB↓	No report	[49]
Obesity	15/15	Unknown	Curcumin C3 complex capsules	100%	1000 mg/d	4 weeks	IL-1β↓, IL-4↓, VEGF↓	No report	[50]
Obesity	20/20	40/0	Curcumin	70%	500 mg 750 mg	12 weeks	ox-LDL↓	No report	[51]
Overweight/obesity	30/30	0/60	Curcumin capsules	100%	500 mg/d	10weeks	IL-6↓, TAC↑, MDA↓	No side effects	[52]
Obesity with NAFLD	42/42	46/38	Nano-curcumin capsules	100%	40 mg bid	3 months	HDL↑, QUICKI↑, WC↓, ALT↓, AST↓, TC↓, LDL-C↓, TG↓, FBG↓, HbA1c↓, TNF-α↓, IL-6↓, hs-CRP↓, HOMA-IR↓, NAFLD severity↓	1 patient had nausea	[53]
Obesity with NAFLD	42/42	46/38	Nano-curcumin capsules	100%	40 mg bid	3 months	BW↓, BMI↓, appetite↓	No report	[54]
Obesity or overweight with migraine	22/22	2/42	Nano-curcumin capsules	100%	40 mg bid	2 months	MCP-1↓	No report	[55]
Obesity with prediabetes	21/20	13/28	Curcumin capsules	95%	500 mg qd	90 days	FBG↓, 2hpp↓, HbA1c↓, FINS↓, HOMA-IR↓, ISI↑	A few subjects had mild nausea, headaches and dizziness	[56]
Overweight with MetS	22/22	17/27	Curcumin tablets	95%	400 mg bid	30 days	BMI↓, WC↓, BW↓, body fat percent↓	No side effects	[57]
NAFLD	37/40	38/39	Curcumin (amorphous dispersion preparation)	14%	500 mg/d	8 weeks	NAFLD severity↓, liver fat content↓, BW↓, BMI↓, TC↓, LDL-C↓, TG↓, HbA1c↓, AST↓, ALT↓	1 patient had stomachache and 2 patients had stomachache and nausea.	[58]
NAFLD	44/43	51/36	Phospholipid curcumin capsules	20%	500 mg bid	8 weeks	TC↓, LDL-C↓, Non-HDL-C↓, TG↓, SUA↓	No side effects	[59]
NAFLD	32/29	37/24	Phospholipid curcumin capsules	20%	250 mg qd	8 weeks	HDL-C ↑, adiponectin↑, leptin↓	No side effects	[60]
NAFLD	22/22	29/15	Phospholipid curcumin capsules	20%	250 mg qd	8 weeks	8-OHdG↓, BMI↓, CML↓, BW↓, WC↓, ALT↓, AST↓, body fat percent↓	No side effects	[61]
NAFLD	23/26	Unknown	Curcumin C3 complex capsules	100%	500 mg qd	8 weeks	NAFLD severity↓, BW↓, TNF-α↓, MCP-1↓	1 patient had digestive problems	[62]
NAFLD	22/23	26/19	Phospholipid curcumin capsules	20%	250 mg qd	8 weeks	BMI↓	No side effects	[63]
NAFLD	35/35	31/39	Curcumin C3 complex capsules	100%	500 mg qd	12 weeks	NAFLD severity↓	No report	[64]
NAFLD	27/23	27/23	Curcumin capsules	95%	500 mg tid	12 weeks	-	No side effects	[65]
NAFLD	35/37	41/31	Phospholipid curcumin capsules	20%	250 mg qd	2 months	NAFLD severity↓, AST↓	No report	[66]
NAFLD	44/43	36/51	Phospholipid curcumin capsules	20%	500 mg bid	8 weeks	BMI↓, WC↓, AST↓, ALT↓, NAFLD severity↓	5 patients had stomachache and flatulence	[5]
NAFLD	36	19/17	Phospholipid curcumin capsules	20%	1500 mg/d	8 weeks	BMI↓, ALT↓, AST↓, LDL-C↓, TG↓, Non-HDL-C↓, SUA↓, NAFLD severity↓	1 patient had nausea	[67]

↑ indicates increase and ↓ indicates decrease. T2DM: type 2 diabetes mellitus; NAFLD: non-alcoholic fatty liver disease; CHD: coronary heart disease; MetS: metabolic syndrome; IGT: impaired glucose tolerance; FBG: fasting blood glucose; HbA1c: hemoglobin A1c; TC: total cholesterol; TG: triglycerides; FINS: fasting insulin; LDL-C: low density lipoprotein cholesterol; HDL-C: high density lipoprotein cholesterol; Non-HDL-C: non-high density lipoprotein cholesterol; VLDL-C: very-low-density lipoprotein cholesterol; HDL: high density lipoprotein; HOMA-β: homeostasis model assessment β; HOMA-IR: homeostasis model assessment-insulin resistance; ISI: insulin sensitivity index; QUICKI: Quantitative Insulin Sensitivity Check Index; ALT: alanine aminotransferase; AST: aspartate aminotransferase; 8-OHdG: 8-Hydroxy-2′-deoxyguanosine; CML: carboxy methyl lysine; BMI: body mass index; BW: body weight; WC: waist circumference; TNF-α: tumor necrosis factor-α; FFA: free fatty acid; CRP: C-reactive protein; hs-CRP: high sensitivity C-reactive protein; IL-6: interleukin-6; SOD: superoxide dismutase; MDA: malonaldehyde; LP(a): lipoprotein(a); MCP-1: macrophage chemoattractant protein-1; GSH: glutathione; TAC: total antioxidant capacity; PAB: pro-oxidant–antioxidant balance; PWV: pulse wave velocity; SUA: serum uric acid; γ-GTP: γ-glutamyl transpeptidase; 2hpp: 2-h postprandial glucose; IL-1β: interleukin-1β; IL-4: interleukin-4; VEGF: vascular endothelial growth factor; ox-LDL: oxidized low density lipoprotein; LPL: lipoprotein lipase; PPAR-γ: peroxisome proliferator-activated receptor; GOT: glutamic oxaloacetic transaminase; GPT: glutamate pyruvate transaminase.

## Data Availability

Not applicable.

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
