# Peer review of "Therapeutic Effect of Curcumin on Metabolic Diseases: Evidence from Clinical Studies"

_ijms, 2023, doi:10.3390/ijms24043323_

Round 1

Reviewer 1 Report

Manuscript is well written. I recommend acceptance.

Author Response

Thank you for your positive comments on our article.

Reviewer 2 Report

Zeng Y et al. demonstrated that curcumin, a natural polyphenolic compound, has the therapeutic effect on metabolic disease including T2DM, obese, NAFLD. It is a very interesting study. However, I concerned about some details.

1.  As you mentioned, there are some types of main plant sources (Fig. 1) as well as various methods for extracting curcumin including extraction. Additionally, purity and exact dose of curcumin are important considerations for evaluating the impact of curcumin. However, there are no description of the manuscript.

2. The delivery forms of curcumin, such as capsule, tablet even powder are vary. What do you think of the best delivery forms of curcumin. The dissolution of curcumin has a possibility of effect on gastric or pancreatic juice. Would you are mentioned about the metabolism of curcumin in stomach?

3. The repetitive pharases are as follow in conclusion section. Modification or simplifications are required.

“The prevalence of metabolic diseases has been rising rapidly worldwide in recent years due to sedentary lifestyles and unhealthy diets. Therefore, it is of great significance to find effective and low toxicity drugs to treat metabolic diseases from natural products. Curcumin has been consumed as a dietary compound for centuries, and its extensive bi-ological activities have been well tested. In recent years, curcumin has been widely con-cerned by the scientific community because of its great potential and application prospect in the treatment of metabolic diseases.”

Thank you.

Author Response

Response to Reviewer 2 Comments:

Zeng Y et al. demonstrated that curcumin, a natural polyphenolic compound, has the therapeutic effect on metabolic disease including T2DM, obese, NAFLD. It is a very interesting study. However, I concerned about some details.

Point 1. As you mentioned, there are some types of main plant sources (Fig. 1) as well as various methods for extracting curcumin including extraction. Additionally, purity and exact dose of curcumin are important considerations for evaluating the impact of curcumin. However, there are no description of the manuscript.

Response 1: Thanks for your opinions. The purity of curcumin has been added to the column 5 of Table 2. The exact dose is shown in column 6.

Point 2. The delivery forms of curcumin, such as capsule, tablet even powder are vary. What do you think of the best delivery forms of curcumin. The dissolution of curcumin has a possibility of effect on gastric or pancreatic juice. Would you are mentioned about the metabolism of curcumin in stomach?

Response 2: Thanks for your opinions. At present, curcumin is administered in various delivery forms, such as curcumin capsules, curcumin tablets and nano-curcumin capsules. Among them, we think that the novel nano-curcumin formulation has a good clinical application prospect, because it can improve the oral bioavailability of curcumin. Based on your question, we have added a paragraph to the " Conclusion and future prospects" to discuss this. Moreover, we have added a paragraph of text to introduce the metabolism of curcumin in the body (page 3). Curcumin is metabolized primarily in the gut and liver. There is no report on the metabolism of curcumin in stomach.

Point 3. The repetitive pharases are as follow in conclusion section. Modification or simplifications are required.

“The prevalence of metabolic diseases has been rising rapidly worldwide in recent years due to sedentary lifestyles and unhealthy diets. Therefore, it is of great significance to find effective and low toxicity drugs to treat metabolic diseases from natural products. Curcumin has been consumed as a dietary compound for centuries, and its extensive bi-ological activities have been well tested. In recent years, curcumin has been widely con-cerned by the scientific community because of its great potential and application prospect in the treatment of metabolic diseases.”

Response 3: Thanks for your opinions. We have simplified the text of this paragraph.

Reviewer 3 Report

The manuscript "Therapeutic effect of curcumin on metabolic diseases: Evidence from clinical studies" is technically a good and correct review paper. It can be accepted.

Author Response

(The authors gave the same response as above.)

Reviewer 4 Report

It is very important to find drugs that can effectively treat metabolic diseases such as 2DM, obesity and NAFLD.

The authors concluded that curcumin may be an effective drug for the treatment of T2DM, obesity and NAFLD. This article is well written and of clinical interest.

However, several issues should be improved before the consideration for publication.

Major comments

1 Many readers are unfamiliar to the curcumin. Therefore, the absorption, distribution, metabolism, and exclusion of curcumin in the body should be described preferably with a figure.

2 What are plausible mechanisms for improving metabolic diseases in the cellular and molecular levels?

  Although the authors described that curcumin decreased levels of the inflammatory factors including TNF-α, IL-6, hs-CRP, MCP-1 and leptin, while increasing levels of the anti-inflammatory marker of adiponectin, where and how to improve these inflammatory factors are unclear in the article.

3 The contents of Figure 3, Figure 4, and Figure 5 seem to be the same. More detail mechanisms are needed to explain the improvement of inflammatory markers and metabolic factors in figures.

4 Are there any sex- and age-related differences in the effects of curcumin?

5 It may be helpful to add the rough cost of curcumin.

Author Response

Response to Reviewer 4 Comments:

It is very important to find drugs that can effectively treat metabolic diseases such as 2DM, obesity and NAFLD. The authors concluded that curcumin may be an effective drug for the treatment of T2DM, obesity and NAFLD. This article is well written and of clinical interest. However, several issues should be improved before the consideration for publication.

Point 1. Many readers are unfamiliar to the curcumin. Therefore, the absorption, distribution, metabolism, and exclusion of curcumin in the body should be described preferably with a figure.

Response 1: Thanks for your opinions. We have added a paragraph of text to introduce the absorption, distribution, metabolism, and exclusion of curcumin in the body (page 3).

Point 2. What are plausible mechanisms for improving metabolic diseases in the cellular and molecular levels? Although the authors described that curcumin decreased levels of the inflammatory factors including TNF-α, IL-6, hs-CRP, MCP-1 and leptin, while increasing levels of the anti-inflammatory marker of adiponectin, where and how to improve these inflammatory factors are unclear in the article.

Response 2: Thanks for your opinions. In this review, our objective was to summarize the clinical evidence of curcumin in the treatment of metabolic diseases. Therefore, the results of animal experiments are not included in this review. So far, clinical trials have not revealed the mechanism of action of curcumin in humans at the molecular and cellular level, possibly due to the difficulty in obtaining tissues and organs. Therefore, we are unable to provide this information. In view of this deficiency, we have a brief discussion in the "Conclusion and future prospects" section.

Point 3. The contents of Figure 3, Figure 4, and Figure 5 seem to be the same. More detail mechanisms are needed to explain the improvement of inflammatory markers and metabolic factors in figures.

Response 3: Thanks for your opinions. Same reply as "Point 2" above. So far, clinical trials have not revealed the mechanism of action of curcumin in humans at the molecular and cellular level. Therefore, the detail mechanisms cannot be provided in the figures.

Point 4. Are there any sex- and age-related differences in the effects of curcumin?

Response 4: Thanks for your opinions. So far, no age-related differences in the effects of curcumin have been reported. However, the clinical efficacy of curcumin is related to gender. Two clinical trials showed that curcumin significantly reduced glucose and LDL-C levels in male subjects with T2DM, while no significant changes were observed in female subjects with T2DM. Based on your question, we have added these in the " Conclusion and future prospects".

Point 5. It may be helpful to add the rough cost of curcumin.

Response 5: Thanks for your opinions. We have added the rough cost of curcumin on page 2.

Reviewer 5 Report

The paper entitled "Therapeutic effect of curcumin on metabolic diseases: Evidence from clinical studies" by Zeng et al. is interesting, however, I have some serious questions regarding the data interpretation
How did the author choose the study, which keywords were used during the searching of the data, from which websites, etc.?

For example, the author mentioned doses in the clinical trials heading, but it's unclear whether the curcumin is used as a single molecule or a sample containing it; if it's a single molecule, please justify how many doses were given at what time interval because regular consumption of a single molecule like curcumin in large quantities may cause toxicity or side effects.

The sample size in the published trials is very hard to understand. Are the authors making claims about the female and male subjects?
In the text, the author just mentions the table results like a report without mentioning any details like mechanism of action, sample size (number of males, females), parameters that were checked in the study, whether the sample is in drug form or some sort of medicinal plant is used, etc.

Please check the grammar and syntax errors as well.

In the conclusion section, the author discusses the previous studies' beneficial effects. The benefits of the current study to the scientific community and the key points of the current study that are not well addressed. Authors, in my opinion, require extensive revision to address the comments thoroughly. 

Author Response

Response to Reviewer 5 Comments:

The paper entitled "Therapeutic effect of curcumin on metabolic diseases: Evidence from clinical studies" by Zeng et al. is interesting, however, I have some serious questions regarding the data interpretation.

Point 1. How did the author choose the study, which keywords were used during the searching of the data, from which websites, etc.?

Response 1: Thanks for your opinions. We have added a paragraph (page 2) to introduce our method of references search.

Point 2. For example, the author mentioned doses in the clinical trials heading, but it's unclear whether the curcumin is used as a single molecule or a sample containing it; if it's a single molecule, please justify how many doses were given at what time interval because regular consumption of a single molecule like curcumin in large quantities may cause toxicity or side effects.

Response 2: Thanks for your opinions. We have added relevant information in Table 2. The purity of the column 5 of Table 2 indicates whether curcumin is used as a single molecule. Moreover, the dosage used (column 6) is also revised and the interval time is indicated.

Point 3. The sample size in the published trials is very hard to understand. Are the authors making claims about the female and male subjects?

Response 3: Thanks for your opinions. The second column of Table 2 (Sample size) shows the numbers of test and control groups respectively. For example, for the first row, 35/35 means 35 subjects in the nano-curcumin group and 35 subjects in the placebo group. According to your opinion, we've indicated the number of female and male subjects in the column 3 of Table 2.

Point 4. In the text, the author just mentions the table results like a report without mentioning any details like mechanism of action, sample size (number of males, females), parameters that were checked in the study, whether the sample is in drug form or some sort of medicinal plant is used, etc.

Response 4: Thanks for your opinions. The text mainly summarizes the results of clinical trials according to disease classification, so only important findings are described, while more detailed information is presented in Table 2. According to your opinion, we have added some relevant information to the text. So far, clinical trials have not revealed the mechanism of action of curcumin in humans at the molecular and cellular level, possibly due to the difficulty in obtaining tissues and organs. Therefore, the detailed molecular mechanism of curcumin cannot be provided at present.

Point 5. Please check the grammar and syntax errors as well.

Response 5: Thanks for your opinions. We have revised the grammar and syntax errors and hope that the language is now acceptable.

Point 6. In the conclusion section, the author discusses the previous studies' beneficial effects. The benefits of the current study to the scientific community and the key points of the current study that are not well addressed. Authors, in my opinion, require extensive revision to address the comments thoroughly.

Response 6: Thanks for your opinions. According to your comments above, we have made extensive revisions to the manuscript. In addition, we have revised the conclusion part based on your and other reviewers' comments.

Round 2

Reviewer 2 Report

I think the authors submitted meticulous responses and corrections.

Thank you for getting it resolved my requirement.  I don't have any additional comments or requests.

Thank you for your meticulous responses and corrections. 

Reviewer 4 Report

I confirmed the manuscript has been improved according to the comments.
This is a good Review.
The manuscript is acceptable now.

Nakajima

Reviewer 5 Report

The current paper is okay for the publication